depression; students; Pakistan; prevalence; mental health; awareness

**Corresponding author:**
Muhammad Moazzam;
Emails: research.associate2@giu.edu.pk,
mmoazzam739@gmail.com

# Prevalence and epidemiology of depression symptoms among Pakistani students: a systematic review and meta-analysis (2000–2025)

Yasmeen Niazi[1], Muhammad Moazzam[1] 🟢, Muhammad Farrukh Asif[1] and Syed Muhammad Yousaf Farooq[1,2]

[1]Office of Research, Innovation and Commercialization, Green International University, Pakistan and [2]Department of Radiography and Medical Imaging Technology, Green International University, Pakistan

## Abstract

This systematic review and meta-analysis was a study that enquired into the prevalence and epidemiology of depression in university students in Pakistan, between 2000 and 2025. Depression is a significant global mental illness with high prevalence in young adulthood. University students are the most susceptible to this risk because of the factors related to it, i.e., academic stress, financial hardships, social pressure, and cultural stigma of mental illness. Although the concerns have been on the increase, the prevalence rates of depression have been widely varied among Pakistani students, with some studies reporting as low as 2.5% to as high as 85%, primarily because of the sampling techniques, assessment instruments, and geographical settings. The present review is based on the findings of 35 studies involving over 11,000 students and suggests that the prevalence rate is approximately 51% in a pooled form, meaning that about 50% of university students in Pakistan are subjected to depressive symptoms. The high level of heterogeneity of the selected studies highlights the acute necessity of the formulation of a standard-based diagnostic criteria and culturally competent mental health assessment instruments. Moreover, systemic challenges, such as the shortage of trained mental health professionals and the general unawareness of the disorder, are continuing to affect the diagnosis and treatment of the disorder at an early stage. According to the results, the necessity of a multi-faceted approach toward mental health, including the establishment of counseling facilities in universities, the development of stress management training, and the federal stigma-reduction campaign, is pressing. The most significant elements of enhancing the well-being of students and the mental health landscape of Pakistan as a whole are early intervention and empowering mental health infrastructure.

## Impact statement

This systematic review and meta-analysis address a critical gap in understanding the mental health challenges faced by university students in Pakistan, a group exposed to distinctive socio-cultural stressors and systemic barriers to care. By synthesizing evidence from 35 studies, this research provides a comprehensive picture of how academic pressure, limited social support, and financial or family-related stress contribute to high rates of depression among students. These challenges are intensified by persistent stigma surrounding mental illness, limited availability of mental health professionals, and the absence of structured psychological support services within universities. The findings have significant implications for public health policy, higher education, and clinical practice. They call for immediate action by policymakers and educational authorities to establish accessible, culturally sensitive mental health programs within universities. Interventions such as on-campus counseling services, peer-led support groups, awareness campaigns, and faculty training in recognizing psychological distress can help reduce stigma and encourage help-seeking behavior.

At the national level, the study supports the integration of standardized mental health screening protocols into university health systems and the inclusion of mental health promotion in education policy frameworks. For mental health professionals, these results emphasize the importance of contextually relevant therapeutic approaches that consider local cultural norms and barriers to disclosure. Globally, this study contributes to the growing recognition of student mental health as a universal concern, particularly in low- and middle-income countries where resources are limited. By highlighting the need for culturally tailored prevention and intervention strategies, this research strengthens the international movement toward equitable, accessible, and sustainable mental health care for young adults in academic settings.

## Introduction

Depressive disorders, often known as depression, are one of the most commonly encountered mental health issues, and the major cause of disability worldwide (GBD, 2019; Mental Disorders Collaborators, 2022; Marx et al., 2023). According to the Diagnostic and Statistical Manual of Mental Disorders, Fifth Edition (DSM-5) and the International Classification of Diseases, 11th Revision (ICD-11), depression is defined as a mood disorder characterized by the presence of a persistent depressed mood or loss of interest or pleasure in most activities for at least 2 weeks, accompanied by additional symptoms such as fatigue, diminished concentration, feelings of worthlessness or guilt, sleep and appetite disturbances, and recurrent thoughts of death or suicide (Ma, 2022; Kogan et al., 2021). It is considered a multi-dimensional problem, leading to impairments in social, occupational, and inter-personal functioning (Schelleman-Offermans et al., 2024). Globally, more than 300 million individuals are estimated to experience depression, representing approximately 4.4% of the world's population. It is also considered one of the major contributors to the overall global disease burden (Liu et al., 2024; Yan et al., 2024).

The transition from adolescence to adulthood is marked as a period of rapid physical, emotional, and psychological changes, placing students at a high risk for mental health issues (Bhattarai et al., 2020). Depression often begins during adolescence and is characterized by a high risk of recurrence. Following the first depressive episode, approximately 40–70% of adolescents experience another episode within 3–5 years (Grossberg and Rice, 2023). The early onset of depression not only manifests typical emotional symptoms but also predisposes individuals to long-term functional decline, including persistent academic difficulties and chronic occupational impairment (Almeida Prado & Santos, De Almeida Prado and Santos, 2022). The cost of affective disorders could be particularly high for young people as their mental well-being is directly linked to societal development and the future workforce productivity (Sajjad et al., 2021).

Despite the reports of a steady rise in its prevalence among students, this population has received comparatively less attention than other groups, such as adolescents in general and older adults. Studies have outlined wide variations in the frequency of students identified as depressed from a relatively small percentage of about 27% (Alshahrani et al., 2024), to high rates 76% (Mao et al., 2019). This broad heterogeneity appears to be affected by numerous factors such as methods of assessment (Moreno-Agostino et al., 2021; Witteveen et al., 2023), geographical location (Kuper et al., 2025), and demographic factors such as socioeconomic status (Farhane-Medina et al., 2022; Mac-Ginty et al., 2024).

Students, particularly those in high-pressure academic environments, are more vulnerable to depression, resulting in poor academic performance, impaired social functioning, poor work performance, and an increased risk of self-harm or suicide (Capdevila-Gaudens et al., 2021; Wallace and Milev, 2021). While some argue that university students are more likely to be advanced socio-economically, which is considered a protective element against depression, there are a number of factors that make students more prone to depression, including low social support, family and relationship issues, financial constraints, adjustment to a new environment, and lifestyle changes resulting in disturbed eating and sleep patterns (Zafar et al., 2017; Al-Azzam et al., 2021).

Overall, the education system in Pakistan faces multiple challenges, including overcrowded classrooms, obsolete curricula, a shortage of teaching staff, and a lack of psychological support services, which can increase the likelihood of depression among students (Khan et al., 2021). Pakistan is also struggling with a critical shortage of mental health professionals. A survey assessing the mental healthcare system in Pakistan was reported to have less than 500 psychiatrists to cater to more than 220 million people, with only 11 psychiatric hospitals and 100 clinical psychologists (Dayani et al., 2024). Depression is estimated to affect 22% of young adults in Pakistan (Ahmed et al., 2016) to 64% (Dawood et al., 2020), with 47% as the average in Karachi (Nisar et al., 2019). Such high levels of depression are attributed to the absence of mental health literacy. Moreover, biomedical factors of the mental disorders are largely neglected in Pakistan because they are regarded as natural outcomes of the stressful life situations (Mashhood et al., 2025). High academic competition and professional insecurities are other crucial variables that cause psychological discomfort among Pakistani students (Jamali et al., 2024). Because of numerous differences in cultures, treatment methods such as psychotherapy, where the patient would have to have elaborate conversations with the therapists, are thus limited in their effectiveness. Since these are the most crucial years in life with regard to academic achievements, career development, and social relations, the impact of untreated depression is especially disastrous (Afu et al., 2023; Deng et al., 2022).

Though different primary research studies have been carried on to estimate the prevalence of depression in Pakistani students, their results differ significantly because of the differences in the methodology, demographic features, and the tools that they use to measure. Such a lack of detailed reviews complicates the interpretation of disjointed findings by researchers and policymakers and the creation of evidence-based interventions. This inconsistency restricts the development of context-driven approaches to the enhancement of mental health. Hence, the proposed research seeks to determine an approximation of the combined prevalence of depression among Pakistani students with the objective of informing future policies and mental health-focused interventions.

## Methodology

### *Study design and guidelines*

The present study is a systematic review and meta-analysis, which was done in accordance with the Preferred Reporting Items of Systematic Reviews and meta-analyses (PRISMA) guidelines (Figure 1) (Liberati et al., 2009), and the purpose of which was to investigate the prevalence and epidemiology of depression among university students in Pakistan between 2000 and 2025.

### *Eligibility criteria (PICO framework)*

To conduct a transparent study selection, the PICO (Population, Intervention/Exposure, Comparator, Outcome) model was applied to develop eligibility criteria.

Population: The studies were eligible based on the participants population; they had to include students of a university in Pakistan in a public or a private institution of higher learning. There were no age, gender, academic program, or year of study limitations. The criteria to include only mixed populations were limited to those studies that reported data specific to university students separately. Exposure: Depression or depressive symptoms were of interest, and any type of screening instrument was used, either a validated or

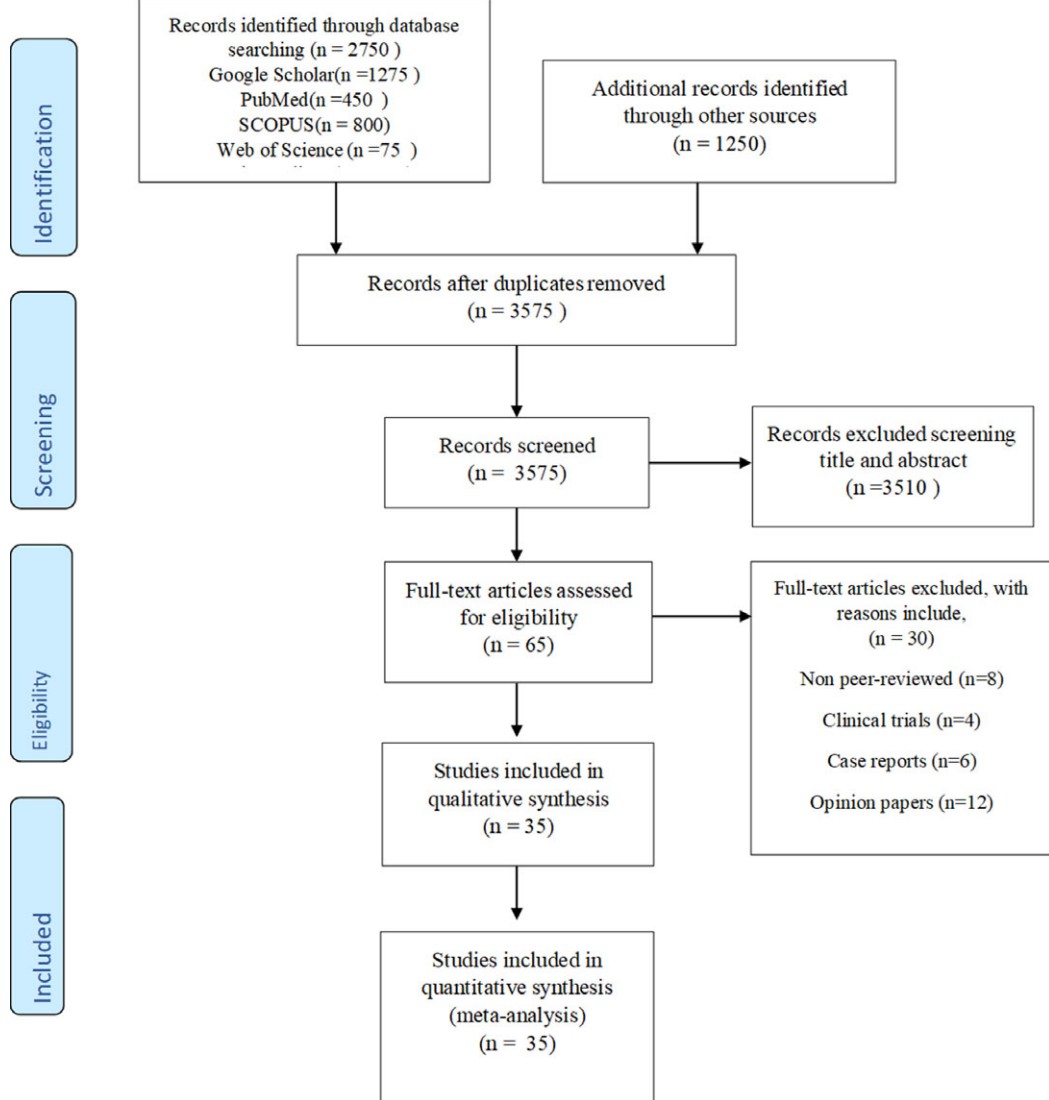

**Figure 1.** PRISMA diagram for selection process.

non-validated scale like PHQ-9, HADS, DASS-21, or BDI. There were no restrictions on cut-off thresholds or diagnostic criteria, although they needed to report a prevalence estimate. **Comparator:** A comparator group was not required, as the purpose of the review was to estimate prevalence rather than compare groups. **Outcome:** The primary outcome was the reported prevalence of depression or depressive symptoms among university students. Eligible studies were those that provided a point or period prevalence, or sufficient quantitative information to calculate a prevalence value. **Study Design:** Eligible study designs included peer-reviewed cross-sectional studies and baseline data from longitudinal studies that reported prevalence estimates relevant to the review question.

Exclusion Criteria: Studies were excluded if they did not meet the above criteria, including experimental studies, clinical trials, qualitative research, mixed-methods studies without extractable quantitative data, case reports, case series, review articles, editorials, commentaries, letters, and conference abstracts. Additional exclusions included studies conducted outside Pakistan, studies involving non-university populations without separate reporting, and non-peer-reviewed or unpublished works.

### Search strategy

A systematic literature search was carried out using Google Scholar, PubMed, SCOPUS, Web of Science and Science Direct between March to June, 2025. The search terms used were: ("depression AND Pakistani students") OR ("depressive students AND Pakistan") (Mental disorders "OR " Students OR Pakistan"). The search was limited to articles published in English between January 2000 and June 2025, as most academic research in Pakistan is disseminated in English. Additionally, reference lists of relevant studies were manually screened to identify further eligible studies. A total of 35 studies met the inclusion criteria and were selected for final analysis after removing duplicates and screening titles, abstracts, and full texts.

### Study selection and data extraction

The titles and abstracts of retrieved studies were screened by two reviewers who independently evaluated the eligibility. The potentially relevant articles were then assessed using the full texts. Few

**Table 1.** Basic characteristics of included studies

| Author (year) | City | Sampling method | Medical/non-medical/both | Age (years ± mean) | Sample Size (n) | Prevalence (n) | Assessment tool | Assessment tool cut-off value |
|---|---|---|---|---|---|---|---|---|
| Alvi et al. (2010) | Wah Cantt | Random sampling | Medical students | 21.4 ± 1.41 | 279 | 98 | Beck Depression Inventory (BDI) | ≥14 |
| Nadeem et al. (2023) | Karachi | Convenience sampling | Medical students | 20.6 ± 1.9 | 79 | 11 | DASS–21 (Depression scale) | ≥10 |
| Inam et al. (2003) | Karachi | Convenience sampling | Medical students | 22.6 ± 1.9 | 189 | 113 | Aga Khan University Anxiety and Depression Scale (AKUADS) | ≥19 |
| Khan et al. (2006) | Karachi | Random sampling | Medical students | 21.3 ± 1.88 | 142 | 99 | Aga Khan University Anxiety and Depression Scale (AKUADS) | >21 |
| Rab et al. (2008) | Lahore | Random sampling | Medical students | 20.7 ± 1.9 | 87 | 17 | Hospital Anxiety and Depression Scale (HADS) | ≥ 8 |
| Hashmi et al. (2014) | Multiple cities | Convenience sampling | Medical students | 21 ± 1.9 | 437 | 199 | Aga Khan University Anxiety and Depression Scale (AKUADS) | ≥20 |
| Zaidi et al. (2023) | Karachi | Random sampling | Medical students | 20.9 ± 2.1 | 284 | 7 | Patient Health Questionnaire–9 (PHQ–9) | ≥10 |
| Nagi et al. (2016) | Lahore | Random sampling | Medical students | 21.1 ± 1.4 | 190 | 122 | Goldberg's Depression Scale | ≥22 |
| Kumar et al. (2019) | Karachi | Purposive sampling | Medical students | 22.74 ± 1.52 | 312 | 180 | DASS–21 (Depression scale) | ≥14 |
| Zafar et al. (2020) | Lahore | Cross-sectional sampling | Medical students | 21 ± 1.9 | 533 | 399 | Patient Health Questionnaire–9 (PHQ–9) | ≥5 |
| Syed et al. (2018) | Multiple cities | Purposive sampling | Medical students | 19.34 ± 1.19 | 267 | 128 | DASS–42 (Depression subscale) | ≥16 |
| Asif et al. (2020) | Sialkot | Random sampling | Medical students | 19.34 ± 1.19 | 500 | 375 | DASS–21 (Depression scale) | ≥10 |
| Azim and Baig (2019) | Karachi | Convenience sampling | Medical students | 21.4 ± 2.2 | 188 | 134 | DASS–21 (Depression subscale) | ≥10 |
| Kim et al. (2023) | Multiple cities | Convenience sampling | Both (proportion not reported) | 21.3 ± 1.88 | 749 | 534 | DASS–21 (Depression subscale) | ≥10 |
| Mansoor et al. (2022) | Lahore | Random sampling | Medical students | 22.74 ± 1.52 | 400 | 191 | DASS–21 (Depression scale) | ≥10 |
| Abrar et al. (2014) | Islamabad | Convenience sampling | Medical students | 21.4 ± 2.2 | 361 | 143 | Aga Khan University Anxiety and Depression Scale (AKUADS) | ≥20 |
| Bukhari and Khanam (2015) | Karachi | Purposive sampling | Nonmedical students | 21.70 ± 2.7 | 331 | 95 | Center for Epidemiological Studies Depression Scale (CES-D) | ≥16 |
| Lail et al. (2021) | Sahiwal | Cross-sectional sampling | Medical students | 19.34 ± 1.19 | 209 | 101 | DASS–42 (Depression, Anxiety, Stress Scale) | ≥10 |
| Rizvi et al. (2015) | Islamabad | Purposive sampling | Medical students | 22.15 ± 1.304 | 66 | 27 | DASS–42 (Depression, Anxiety, Stress Scale) | ≥10 |
| Ali et al. (2014) | Karachi | Convenience sampling | Nonmedical students | 20.36 ± 1.58 | 557 | 228 | Aga Khan University Anxiety and Depression Scale (AKUADS) | ≥19 |
| Ul Haq et al. (2018) | Lahore | Convenience sampling | Both (proportion not reported) | 19.34 ± 1.19 | 361 | 132 | DASS–21 (Depression scale) | ≥10 |
| Naz et al. (2017) | Lahore | Cross-sectional sampling | Medical students | 19.34 ± 1.19 | 152 | 47 | DASS–21 (Depression scale) | ≥10 |

*(Continued)*

**Table 1.** (*Continued*)

| Author (year) | City | Sampling method | Medical/non-medical/both | Age (years ± mean) | Sample Size (n) | Prevalence (n) | Assessment tool | Assessment tool cut-off value |
|---|---|---|---|---|---|---|---|---|
| Aqeel et al. (2022) | Multiple cities | Cross-sectional sampling | Both (proportion not reported) | 20.31 ± 1.51 | 500 | 125 | Beck Depression Inventory (BDI-II) | ≥14 |
| Abbas et al. (2015) | Multiple cities | Cross-sectional sampling | Nonmedical students | 21.70 ± 2.7 | 433 | 270 | Standardized questionnaire based on Bell et al. (2006) | ≥14 |
| Rehan (2022) | Multiple cities | Convenience sampling | Nonmedical students | 22.74 ± 1.52 | 205 | 123 | GAD–7 (Generalized Anxiety Disorder), PHQ–9 | ≥10 |
| Talpur et al. (2023) | Jamshoro | Convenience sampling | Nonmedical students | 20.73 ± 1.24 | 186 | 133 | DASS–21 (Depression scale) | ≥28 |
| Rajar et al. (2022) | MirpurKhas | Random sampling | Medical students | 20.73 ± 1.24 | 186 | 63 | DASS–42 (Depression, Anxiety, Stress Scale) | ≥10 |
| Arif et al. (2021) | Lahore | Cross-sectional sampling | Medical students | 23.55 ± 1.42 | 104 | 28 | DASS–21 (Depression scale) | ≥28 |
| Hayat et al. (2021) | Multiple cities | Convenience sampling | Medical students | 20.73 ± 1.24 | 145 | 65 | Beck Depression Inventory (BDI) | ≥20 |
| Sajjad et al. (2021) | Multan | Random sampling | Medical students | 22.74 ± 1.52 | 315 | 201 | Patient Health Questionnaire–9 (PHQ–9) | ≥10 |
| Shabbir et al. (2022) | Faisalabad | Convenience sampling | Medical students | 25.22 ± 2.19 | 98 | 79 | Beck Depression Inventory (BDI) | ≥10 |
| Ikram et al. (2018) | Karachi | Convenience sampling | Medical students | 21.10 ± 1.931 | 154 | 93 | Center for Epidemiological Studies Depression Scale (CES-D) | ≥15 |
| Gul et al. (2020) | Multiple cities | Random sampling | Both (proportion not reported) | 22.74 ± 1.52 | 1,159 | 986 | Depression screening test | ≥10 |
| Uttra et al. (2017) | Sargodha | Random sampling | Medical students | 22.74 ± 1.52 | 200 | 151 | Patient Health Questionnaire–9 (PHQ–9) | ≥5 |
| Lakhiar et al. (2017) | Nawabshah | Cross-sectional sampling | Nonmedical students | 21.06 ± 1.836 | 851 | 568 | Aga Khan University Anxiety and Depression Scale (AKUADS) | ≥20 |

disputes were also raised concerning studies that had incomplete methodological data or lacked clear diagnostic definitions. These were settled down in a discussion and when there was no agreement, the third reviewer, who specialized in systematic reviews and mental health study, was consulted to decide on them.

A standardized extraction form was used to extract the data. In each study, the following information was taken: Author (Year), City, Sampling Method, Settings (University/Region), Sample Size (n), Age (Mean) years, Prevalence of depression (n), Assessment Tool of depression, and Risk Factors of depression.

### Risk of Bias assessment

The risk of bias was assessed by the Joanna Briggs Institute Critical Appraisal Checklist for Prevalence Studies (Joanna Briggs Institute, 2017). We also used a modified version of the checklist and changed one of the items to determine the validity and reliability of the depression screening scale in the Pakistani setting. The checklist had nine domains that are: sample representativeness, recruitment methods, sample size, description of subjects and settings, validity of the measurement tool and sufficiency of the response rate. They were rated and classified into Low risk of bias (score ≥ 7), Moderate risk (score 4–6) and High risk (score < 4). The possibility of bias was

recorded but was not a factor in determining inclusion in the meta-analysis.

### Data synthesis and statistical analysis

All the statistical analyses were conducted via MedCalc software and R-programming language. The prevalence rates of depressive symptoms were summarized using descriptive statistics in different studies. Since the heterogeneity of the study was high, a random-effects meta-analysis model was used in order to obtain the pooled prevalence and its 95% confidence intervals (CIs). To visualize these pooled estimates, forest plots were produced. The level of heterogeneity was measured based on the Q statistic, $Tau^2$ and $I^2$ statistic, where $I^2 \geq 75\%$ represented high levels of heterogeneity (*p*-value of lower than 0.05).

### Subgroup and Meta-regression analyses

Subgroup analyses were performed to examine possible heterogeneity of the included studies based on various categorical variables. These were: the city in which the study was carried out, sample size, sampling technique, the measure of depression that was used in the screening, the type of university (public and private) and the study

area (medical and non-medical students). In the case of meta-regression, a mixed-effects model was used to investigate the association between pre-specified study-level covariates, such as the year of the study, the size of the sample, and the validity of the assessment tool, and the variation in the depression prevalence estimates. Regression coefficients and heterogeneity of the residual ($\tau^2$) were estimated via restricted maximum likelihood (REML). These subgroup comparisons and meta-regression were conducted by use of a random-effects model. The *p*-value below 0.05 was taken as a sign of a statistically significant difference between subgroups.

## Results

The systematic review involved 35 studies carried out in different cities in Pakistan, and the total sample size was 11,209 participants. The sample sizes were between 66 and 1,159 (median = 279). Most of the studies were carried out on medical students (71.4%), a minority of 14.3% carried out non-medical cohorts and 14.3% carried out mixed cohorts. Research was focused in urban centers geographically, with 25.7% of the study in Karachi, 20% in Lahore and 22.9% crossing cities.

These assessment instruments were diverse, but the most common was the Depression Anxiety Stress Scales (DASS-21 and DASS-42), mostly DASS-21 (40% of studies). The Patient Health Questionnaire-9 (PHQ-9) was used in 14.3% of studies, particularly in recent studies. Other scales were the Aga Khan University Anxiety and Depression Scale (AKUADS), Beck Depression Inventory (BDI/BDI-II) and Depression Scale by Goldberg, as well as CES-D.

The sampling strategies were mainly convenience sampling (34.3%), random sampling (31.4%), purposive sampling (14.3%), and cross-sectional methods (20%). There was a lot of variation in prevalence, with 2.5% being the lowest and 85 the highest. Research on medical students found an average depressive prevalence of around 42.6%, which is a little higher than the non-medical populations at 38.2%. Recent research has shown methodological standardization improvement, with 60.7% employed before 2020 and 71.4% employed across 2020. Geographical analysis revealed that Karachi-based studies frequently utilized AKUADS (17.1%), whereas Lahore and other cities employed DASS-21 and PHQ-9 more commonly, highlighting regional differences in assessment preferences.

### *Prevalence of depression in university students in Pakistan*

Table 2 shows that the meta-analysis of 35 studies that include 11,209 university students in Pakistan provides an estimate of 51% as an overall prevalence of depressive symptoms based on a

**Table 2.** Meta-analysis of prevalence of depressive symptoms 35 studies among university students in Pakistan

| Study | Sample size | Proportion (%) | 95% CI | Weight (%) Fixed | Random |
|---|---|---|---|---|---|
| Alvi et al. (2010) | 279 | 35.125 | 29.530 to 41.041 | 2.49 | 2.87 |
| Nadeem et al. (2023) | 79 | 13.924 | 7.161 to 23.550 | 0.71 | 2.76 |
| Inam et al. (2003)) | 189 | 59.788 | 52.425 to 66.840 | 1.69 | 2.85 |
| Khan et al. (2006) | 142 | 69.718 | 61.452 to 77.141 | 1.27 | 2.83 |
| Rab et al. (2008) | 87 | 19.540 | 11.815 to 29.432 | 0.78 | 2.77 |
| Hashmi et al. (2014) | 437 | 45.538 | 40.798 to 50.338 | 3.90 | 2.89 |
| Zaidi et al. (2023) | 284 | 2.465 | 0.997 to 5.012 | 2.53 | 2.87 |
| Nagi et al. (2016) | 190 | 64.211 | 56.951 to 71.019 | 1.70 | 2.85 |
| Kumar et al. (2019) | 312 | 57.692 | 51.999 to 63.238 | 2.78 | 2.88 |
| Zafar et al. (2020) | 533 | 74.859 | 70.951 to 78.490 | 4.75 | 2.90 |
| Syed et al. (2018) | 267 | 47.940 | 41.813 to 54.114 | 2.38 | 2.87 |
| Asif et al. (2020) | 500 | 75.000 | 70.963 to 78.738 | 4.46 | 2.90 |
| Azim and Baig (2019) | 188 | 71.277 | 64.241 to 77.627 | 1.68 | 2.85 |
| Kim et al. (2023) | 749 | 71.295 | 67.909 to 74.512 | 6.67 | 2.91 |
| Mansoor et al. (2022) | 400 | 47.750 | 42.762 to 52.771 | 3.57 | 2.89 |
| Abrar et al. (2014) | 361 | 39.612 | 34.532 to 44.864 | 3.22 | 2.89 |
| Bukhari and Khanam (2015) | 331 | 28.701 | 23.886 to 33.901 | 2.95 | 2.88 |
| Lail et al. (2021) | 209 | 48.325 | 41.378 to 55.321 | 1.87 | 2.86 |
| Rizvi et al. (2015) | 66 | 40.909 | 28.955 to 53.706 | 0.60 | 2.73 |
| Ali et al. (2014) | 557 | 40.934 | 36.818 to 45.146 | 4.96 | 2.90 |
| Ul Haq et al. (2018) | 361 | 36.565 | 31.587 to 41.766 | 3.22 | 2.89 |
| Naz et al. (2017) | 152 | 30.921 | 23.685 to 38.919 | 1.36 | 2.83 |
| Aqeel et al. (2022) | 500 | 25.000 | 21.262 to 29.037 | 4.46 | 2.90 |

*(Continued)*

**Table 2.** (*Continued*)

| Study | Sample size | Proportion (%) | 95% CI | Weight (%) Fixed | Random |
|---|---|---|---|---|---|
| Abbas et al. (2015) | 433 | 62.356 | 57.605 to 66.936 | 3.86 | 2.89 |
| Rehan (2022) | 205 | 60.000 | 52.945 to 66.761 | 1.83 | 2.86 |
| Talpur et al. (2023) | 186 | 71.505 | 64.441 to 77.870 | 1.66 | 2.85 |
| Rajar et al. (2022) | 186 | 33.871 | 27.108 to 41.158 | 1.66 | 2.85 |
| Arif et al. (2021) | 104 | 26.923 | 18.694 to 36.510 | 0.93 | 2.79 |
| Hayat et al. (2021) | 145 | 44.828 | 36.570 to 53.301 | 1.30 | 2.83 |
| Sajjad et al. (2021) | 315 | 63.810 | 58.234 to 69.123 | 2.81 | 2.88 |
| Shabbir et al. (2022) | 98 | 80.612 | 71.391 to 87.905 | 0.88 | 2.79 |
| Ikram et al. (2018) | 154 | 60.390 | 52.201 to 68.170 | 1.38 | 2.84 |
| Gul et al. (2020) | 1,159 | 85.073 | 82.890 to 87.077 | 10.32 | 2.91 |
| Uttra et al. (2017) | 200 | 75.500 | 68.936 to 81.292 | 1.79 | 2.85 |
| Lakhiar et al. (2017) | 851 | 66.745 | 63.467 to 69.906 | 7.58 | 2.91 |
| Total (fixed effects) | 11,209 | 55.987 | 55.063 to 56.907 | 100.00 | 100.00 |
| Total (random effects) | 11,209 | 50.588 | 43.012 to 58.150 | 100.00 | 100.00 |

random-effects model, in addition to an estimate of 56% with a fixed-effects model. The prevalence of individual studies was very heterogeneous, with a low rate of 2.5% (Zaidi et al., 2023) and as high as 85.07% (Gul et al., 2020). More significant studies, like the study done by Khan et al. (2021), covered a larger sample and therefore their results added more value to the pooled estimate compared to the studies with smaller samples. The individual study confidence intervals also showed different rates of precision, with larger sample sizes having narrower confidence intervals. On the whole, the obtained results show that the depressive symptoms are experienced by about half of the university students' population in Pakistan, indicating a significant burden of mental health that is to be addressed. The difference in prevalence among studies draws attention to the necessity of standardly assessing tools and methods of sampling in future studies (Table 2) (Figure 2).

Table 3 presents a subgroup analysis examining the prevalence of depression among university students based on various study characteristics. The prevalence rates varied across different subgroups, with the highest observed in studies using the DASS-21 (subscale) tool at 71.25%, and the lowest in studies utilizing the CES-D at 44.11%. The prevalence estimates in the sampling methods were as follows: about 43.76% (purposive sampling) and 53.62% (convenience sampling). The analyses of university type revealed a prevalence of about 49.43% at public universities, 46.52% in private institutions and 54.83% in combined private/public. In terms of student type, it was found that medical students had a prevalence of about 48.72%, and non-medical students had a higher prevalence of 60.33%. The urban analysis showed that its prevalence was similar in Islamabad (39.86%), Karachi (43.03%), Lahore (43%), and in numerous cities (55.73%). The heterogeneity of subgroups was great ($I^2 > 94$) in all categories, which means that there is a large difference between studies. All subgroup differences were statistically significant ($p < 0.0001$), suggesting that study characteristics influence the estimated prevalence of depression among university students.

### Meta-regression analysis of factors associated with depression prevalence

A meta-regression was implemented to examine possible sources of heterogeneity that included study year, sample size, and assessment tool. The mixed-effects model demonstrated that 3.87% of between-study variance was explained by moderators ($R^2 = 3.87\%$), and the remainder of the heterogeneity was large ($\tau^2 = 0.0402$; $I^2 = 98.41$; QE (31) = 2,927.15). The general moderator test was not statistically significant (QM(3) = 4.28; $p = 0.233$), which means that the variables did not have a meaningful influence on the heterogeneity. Personal predictors showed a significant but small correlation with the sample size only ($\beta = 0.0003$; $p = 0.039$), whereas the type of assessment tool and study year were not found to be significant predictors. The results indicate that, despite the study characteristics reviewed, the prevalence estimates of depression were very heterogeneous across the included studies (Table 4).

### Assessment of publication Bias

The Egger and Begg tests were conducted to assess the possible presence of publication bias. Egger test indicated the presence of funnel plot asymmetry (intercept = $-9.71$; 95%CI $-17.69$ to $-1.74$; $p = 0.0185$), which was interpreted as a possible publication bias, yet Begg test (Kendall tau($\tau$) = $-0.11$; $p = 0.35$) did not find any bias. To further evaluate and correct the possibility of bias, trim-and-fill analysis was performed. The analysis did not impute any extra studies and the adjusted pooled prevalence (50.6%; 95% CI: 47.0–54.2%) was the same as the unadjusted model, indicating that publication bias did not have a significant effect on the overall estimate. However, heterogeneity levels were significant ($I^2 = 98.4$, $p = 0.001$), which implies that there is much variability among studies, which may be caused by the instrument, cut-off scores, and sample differences. Hence, although publication bias might exist, its impact on the pooled prevalence seems to be minimal, and the overall results must be viewed cautiously (Figure 3).

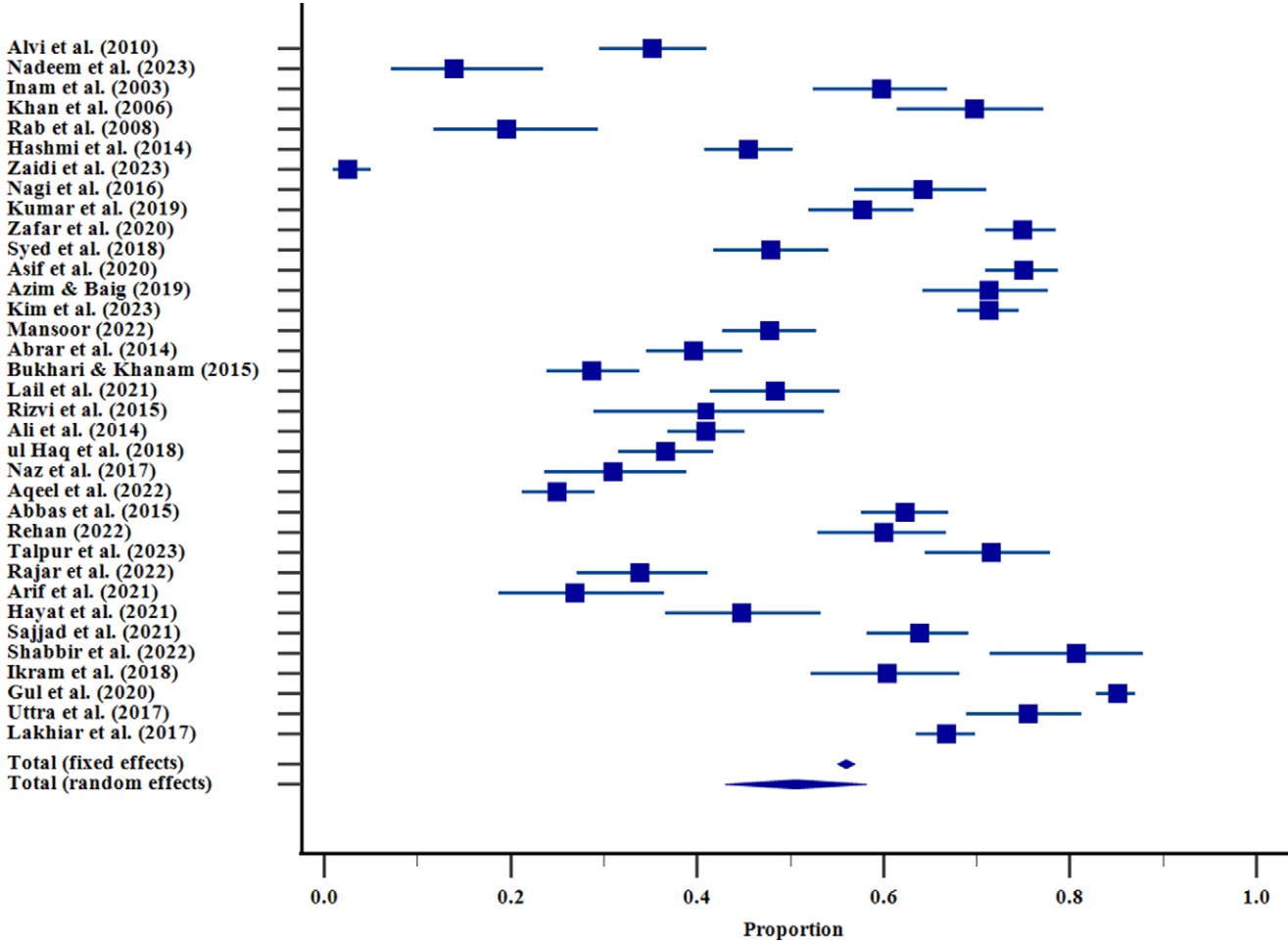

**Figure 2.** Forest plot of combined prevalence of depressive symptoms among Pakistani university students (2000–2025) among 35 included studies and random-effects model. Horizontal lines are 95% confidence intervals, and the diamond represents the pooled estimate in general.

**Table 3.** Subgroup analysis based on study characteristics

| Group | Sub-groups | Number of studies | Prevalence (%) | 95% CI | $I^2$ | Q | df | P |
|---|---|---|---|---|---|---|---|---|
| Sampling method | Convenience | 7 | 56.481 | 45.55–67.09 | 95.29 | 127.3155 | 6 | <0.0001 |
| | Cross-sectional | 17 | 51.888 | 41.61–62.08 | 98.61 | 1,151.0460 | 16 | <0.0001 |
| | Purposive | 3 | 38.940 | 25.70–53.06 | 91.61 | 23.8351 | 2 | <0.0001 |
| | Random | 8 | 46.797 | 25.06–69.18 | 99.04 | 731.8423 | 7 | <0.0001 |
| Sampling technique (random sampling) | Simple random sampling | 4 | 49.602 | 24.08–75.22 | 98.26 | 172.52 | 3 | <0.0001 |
| | Systematic stratified random sampling | 4 | 44.092 | 11.41–80.04 | 99.37 | 473.9140 | 3 | <0.0001 |
| Sampling technique(Cross-sectional studies) | Convenience sampling | 12 | 50.894 | 41.28–60.47 | 97.41 | 424.3656 | 11 | <0.0001 |
| | Stratified random sampling | 2 | 50.173 | 24.47–75.82 | 98.04 | 51.0734 | 1 | <0.0001 |
| | Snowball sampling technique | 2 | 41.885 | 11.73–75.9 | 98.69 | 76.2682 | 1 | <0.0001 |
| Sampling technique(Purposive sampling sampling) | Non-probability purposive | 3 | 38.940 | 25.70–53.06 | 91.61 | 23.8351 | 2 | <0.0001 |
| University Type | Public | 9 | 49.43 | 31.62–67.32 | 98.91 | 733.73 | 8 | <0.0001 |
| | Private | 12 | 46.52 | 35.88–57.33 | 97 | 366.96 | 11 | <0.0001 |
| | Both private/public | 14 | 54.83 | 42.78–66.61 | 98.77 | 1,054.58 | 13 | <0.0001 |

*(Continued)*

**Table 3.** (*Continued*)

| Group | Sub-groups | Number of studies | Prevalence (%) | 95% CI | $I^2$ | Q | df | P |
|---|---|---|---|---|---|---|---|---|
| Type of students | Medical students | 24 | 48.72 | 39.92–57.57 | 97.89 | 1,139.1 | 24 | <0.0001 |
| | Non-medical students | 5 | 60.33 | 49.04–71.08 | 96.37 | 110.08 | 4 | <0.0001 |
| | Both medical/nonmedical students | 5 | 49.88 | 23.96–75.84 | 99.56 | 902.45 | 4 | <0.0001 |
| City | Islamabad | 2 | 39.86 | 35.28–44.53 | 0 | 0.05 | 1 | 0.8297 |
| | Karachi | 9 | 43.03 | 26.04–60.92 | 98.62 | 581.16 | 8 | <0.0001 |
| | Lahore | 7 | 43 | 27.89–58.82 | 97.77 | 269.37 | 6 | <0.0001 |
| | Multiple cities | 8 | 55.73 | 39.21–71.63 | 99.05 | 737.76 | 7 | <0.0001 |
| Depression tool | AKUADS | 6 | 53.67 | 42.43–64.72 | 96.8 | 156.04 | 5 | <0.0001 |
| | BDI | 3 | 53.87 | 28.61–78.12 | 97 | 66.57 | 2 | <0.0001 |
| | CES-D | 2 | 44.11 | 15.86–74.65 | 97.74 | 44.18 | 1 | <0.0001 |
| | DASS–21 (full) | 8 | 45 | 31.10–59.30 | 97.67 | 300.19 | 7 | <0.0001 |
| | DASS–21 (subscale) | 2 | 71.25 | 68.31–74.10 | 0 | 0 | 1 | 0.9779 |
| | DASS–42 | 3 | 41.1 | 31.59–50.96 | 76.51 | 8.51 | 2 | 0.0142 |
| | PHQ–9 | 4 | 51.07 | 14.45–87.02 | 99.54 | 656.99 | 3 | <0.0001 |
| Depression tool (AKUADS cut-off value) | ≥19 | 2 | 50.120 | 32.01–68.20 | 95.06 | 20.2520 | 1 | <0.0001 |
| | ≥20 | 3 | 50.800 | 33.40–68.09 | 97.99 | 99.6163 | 2 | <0.0001 |
| Depression tool (DASS–21 (Depression scale) (cut-off value) | ≥10 | 5 | 40.626 | 21.99–60.78 | 98.31 | 237.3213 | 4 | <0.0001 |
| | ≥28 | 2 | 49.296 | 10.44–88.67 | 98.23 | 56.5297 | 1 | <0.0001 |
| Depression tool (DASS–21 (depression subscale) scale) cut-off value | ≥10 | 2 | 71.246 | 68.31–74.09 | 0.00 | 0.0008 | 1 | 0.9779 |
| Depression tool (DASS–42 scale) cut-off value | ≥10 | 3 | 41.10 | 31.59–50.96 | 76.51 | 8.5135 | 2 | 0.0142 |
| Depression tool (PHQ–9) cut-off value | ≥5 | 2 | 74.97 | 71.77–78.03 | 0.00 | 0.0246 | 1 | 0.8755 |
| | ≥10 | 3 | 37.37 | 2.51–83.79 | 99.51 | 411.7004 | 2 | <0.0001 |

**Table 4.** Meta-regression results examining moderators of depression prevalence among university students

| Variable | Estimate ($\beta$) | SE | 95% CI (Lower–Upper) | z-value | *p*-value |
|---|---|---|---|---|---|
| Study year | −0.0018 | 0.0077 | −0.0169 to 0.0132 | −0.2384 | 0.8116 |
| Sample size | **0.0003** | 0.0001 | 0.0000 to 0.0006 | **2.0636** | **0.0391*** |
| Assessment tool | 0.0027 | 0.0108 | −0.0185 to 0.0240 | 0.2509 | 0.8019 |

*$p < 0.05$.

### Risk of bias in included studies

Most studies demonstrated a low risk of bias, with a total score of 8, indicating robust methodology across key domains such as sample representativeness, sampling technique, depression assessment, measurement, statistical analysis, and response rate. In particular, 25 studies obtained a low-risk score that reflects the quality of methodology.

A few studies showed moderate risk of bias, with total scores of 5 or 6, which is merely because of the non-random sampling or the failure in reporting the details of measurement (Ul Haq et al., 2018; Rehan, 2022; Ikram et al., 2018). Moreover, Kumar et al. (2019) obtained 6 points, which corresponds to moderate concerns connected primarily with the sampling technique (Table 5).

This review study brings forward a lot of risk factors, major findings, and suggested interventions for depression among university students. Some examples are female sex, early-years status, academic demands, social and environmental stress, family psychological conditions, a negative perception of the learning environment (Alvi et al., 2010; Nadeem et al., 2023; Inam et al., 2003); Khan et al., 2006; Rab et al., 2008). All the studies indicate that the prevalence rate of depression is high, and mostly students experience it during the first years of their university life, and females are usually more prone to it (Hashmi et al., 2014; Naz et al., 2017; Sajjad et al., 2021). There is a special appeal to seek out and implement mental support systems, including counseling, stress management

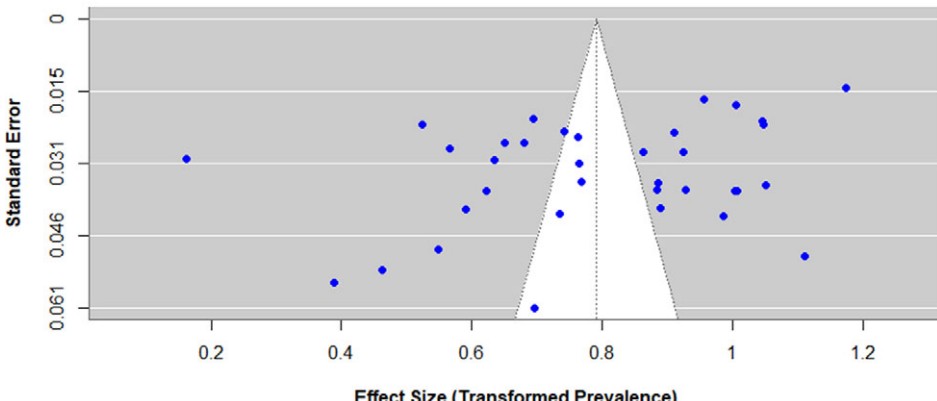

**Figure 3.** Funnel plot that evaluates the risk of publication bias by the studies involving depressive symptoms in Pakistani university students. Individual estimates of prevalence of statistics in the study are represented on the x-axis and the standard errors (SEs) on the y-axis. All the points represent one study; the imbalance in the plot can be evidence of a potential publication bias or heterogeneity between studies.

**Table 5.** Risk of bias in included studies

| Author (Year) | Sample representativeness | Sample technique | Sample size | Subjects settings | Ascertainment of depression | Measurement of the depression | Statistical analysis | Response rate | Total Score | Risk of bias |
|---|---|---|---|---|---|---|---|---|---|---|
| Alvi et al. (2010) | 1 | 1 | 1 | 1 | 1 | 1 | 1 | 1 | 8 | Low |
| Nadeem et al. (2023) | 1 | 1 | 1 | 1 | 1 | 1 | 0 | 1 | 7 | Low |
| Inam et al. (2003) | 1 | 0 | 0 | 1 | 1 | 1 | 0 | 0 | 4 | High |
| Khan et al. (2006) | 1 | 1 | 1 | 1 | 1 | 1 | 1 | 1 | 8 | Low |
| Rab et al. (2008) | 1 | 1 | 1 | 1 | 1 | 1 | 1 | 1 | 8 | Low |
| Hashmi et al. (2014) | 1 | 1 | 1 | 1 | 1 | 1 | 1 | 1 | 8 | Low |
| Zaidi et al. (2023) | 1 | 1 | 1 | 1 | 1 | 1 | 1 | 1 | 8 | Low |
| Nagi et al. (2016) | 1 | 1 | 1 | 1 | 1 | 1 | 1 | 0 | 7 | Low |
| Kumar et al. (2019) | 1 | 0 | 0 | 1 | 1 | 1 | 1 | 1 | 6 | Moderate |
| Zafar et al. (2020) | 1 | 1 | 1 | 1 | 1 | 1 | 1 | 1 | 8 | Low |
| Syed et al. (2018) | 1 | 1 | 1 | 1 | 1 | 1 | 1 | 1 | 8 | Low |
| Asif et al. (2020) | 1 | 1 | 1 | 1 | 1 | 1 | 1 | 1 | 8 | Low |
| Azim and Baig (2019) | 1 | 1 | 1 | 1 | 1 | 1 | 1 | 1 | 8 | Low |
| Kim et al. (2023) | 1 | 1 | 1 | 1 | 1 | 1 | 1 | 1 | 8 | Low |
| Mansoor et al. (2022) | 1 | 1 | 1 | 1 | 1 | 1 | 1 | 1 | 8 | Low |
| Abrar et al. (2014) | 1 | 1 | 1 | 1 | 1 | 1 | 1 | 1 | 8 | Low |
| Bukhari and Khanam (2015) | 1 | 1 | 1 | 1 | 1 | 1 | 1 | 1 | 8 | Low |
| Lail et al. (2021) | 1 | 1 | 1 | 1 | 1 | 1 | 1 | 1 | 8 | Low |
| Rizvi et al. (2015) | 1 | 1 | 0 | 1 | 1 | 1 | 1 | 1 | 7 | Moderate |
| Ali et al. (2014) | 1 | 1 | 1 | 1 | 1 | 1 | 1 | 1 | 8 | Low |
| ul Haq et al. (2018) | 1 | 0 | 0 | 1 | 1 | 1 | 0 | 1 | 5 | Moderate |
| Naz et al. (2017) | 1 | 1 | 1 | 1 | 1 | 1 | 1 | 0 | 7 | Low |
| Aqeel et al. (2022) | 1 | 1 | 1 | 1 | 1 | 1 | 1 | 1 | 8 | Low |
| Abbas et al. (2015) | 1 | 1 | 1 | 1 | 1 | 1 | 1 | 1 | 8 | Low |
| Rehan (2022) | 1 | 0 | 0 | 1 | 1 | 1 | 0 | 1 | 5 | Moderate |
| Talpur et al. (2023) | 1 | 1 | 1 | 1 | 1 | 1 | 1 | 1 | 8 | Low |
| Rajar et al. (2022) | 1 | 1 | 1 | 1 | 1 | 1 | 1 | 1 | 8 | Low |

*(Continued)*

**Table 5.** (*Continued*)

| Author (Year) | Sample representativeness | Sample technique | Sample size | Subjects settings | Ascertainment of depression | Measurement of the depression | Statistical analysis | Response rate | Total Score | Risk of bias |
|---|---|---|---|---|---|---|---|---|---|---|
| Arif et al. (2021) | 1 | 1 | 1 | 1 | 1 | 1 | 1 | 1 | 8 | Low |
| Hayat et al. (2021) | 1 | 1 | 1 | 1 | 1 | 1 | 0 | 1 | 7 | Low |
| Sajjad et al. (2021) | 1 | 1 | 1 | 1 | 1 | 1 | 1 | 1 | 8 | Low |
| Shabbir et al. (2022) | 1 | 1 | 1 | 1 | 1 | 1 | 1 | 1 | 8 | Low |
| Ikram et al. (2018) | 1 | 0 | 0 | 1 | 1 | 1 | 1 | 1 | 6 | Moderate |
| Gul et al. (2020) | 1 | 1 | 1 | 1 | 0 | 1 | 1 | 1 | 7 | Low |
| Uttra et al. (2017) | 1 | 1 | 1 | 1 | 1 | 1 | 1 | 1 | 8 | Low |
| Lakhiar et al. (2017) | 1 | 1 | 1 | 1 | 1 | 1 | 1 | 1 | 8 | Low |

interventions, and awareness interventions (Kumar et al., 2019; Zafar et al., 2020; Syed et al., 2018). Recommendations often involve early-stage screening, psychoeducation, healthy coping skills encouragement, living and academic environment improvements, as well as decreasing the stigma toward people with mental issues (Azim and Baig, 2019; Kim et al., 2023; Rehan, 2022). Specific interventions to reduce or solve concerned risk factors, such as academic overload, social stigma, and environmental stressors, must be used to create effective prevention and management, which will eventually build a healthier mental environment in the university environment (Ali et al., 2014; Uttra et al., 2017; Lakhiar et al., 2017).

## Discussion

The results of this systematic review and meta-analysis highlight an alarming rate of depression among university students in Pakistan, indicating that about half of the students encounter depressive symptoms. The pooled prevalence of depression was estimated to be approximately 51%, which is a very high level of burden with regard to mental health in this group. By contrast, the prevalence rates reported in neighboring South Asian countries by meta-analyses stand at about 50% in India (Dutta et al., 2023), 19.4% in Sri Lanka (Alwis et al., 2024), and 47% in Bangladesh (Hosen et al., 2021). The recent systematic reviews demonstrate that the rate of depression among university students varies between 28.13% and 38.80% throughout the whole world (Mekonnen et al., 2024; Anbesaw et al., 2023; Shorey et al., 2022). The prevalence of 51% in Pakistan thus exceeds the regional as well as the global prevalence, which points to an urgent issue of public health concern. These high prevalence rates align with the trends in other countries and are particularly worrying given the severity of socio-cultural and infrastructural challenges that Pakistani students experience. The significant heterogeneity is represented by wide variance in the estimates of individual studies, ranging between as low as 2.5% and as high as 85%, perhaps because of differences in assessment instruments, sampling methods, regional differences, and demographic factors. This heterogeneity emphasizes the fact that the true burden of depression in this population cannot be adequately measured and that future studies require special methodologies.

This broad discrepancy in prevalence estimates is likely to be strongly correlated with the diverse types of assessment instruments employed in various studies, and each one of them differs in terms of construct focus, scoring thresholds, and cultural appropriateness.

The findings of studies using the DASS-21 have been found to yield significantly greater prevalence, perhaps because of the sensitivity to general psychological distress and the presence of symptoms related to stress, which could overpower depression measures in student samples. Conversely, the PHQ-9, which is more diagnostically driven and conforms to the DSM criteria, was more likely to give smaller estimates, and therefore, provided a more conservative estimate of clinically significant depressive symptoms. Other tools like the AKUADS, which include somatic and anxiety-related items that are culturally specific, could further increase the scores in those settings where psychological distress is usually manifested somatically. These differences in methodology point to the effects of measurement choice on prevalence that has been reported and the importance of increased standardization and culturally validated tools in enhancing comparability and validity in future studies.

The high values of $I^2$ indicated that the studies varied greatly, which implied that there are many factors that are linked to depression prevalence among Pakistani students. The research based on the Depression Anxiety Stress Scales (DASS-21) in particular showed a significantly higher prevalence, which largely exceeded 70%, compared to the research that used different scales, such as CES-D or PHQ-9, which estimated the prevalence rate to be between 44% and 55%. Such differences may be explained by the fact that these measurement tools are sensitive and specific in different ways and that there are culture-specific problem that affect the way the student understands and perceives screening questions. Additionally, with some apparent geographical variations, the prevalence rates were found to be significantly higher in the urban areas, as compared to the rural areas or the small cities, which might be due to greater academic requirements, socio-economic burden and stress related to urban life in the large cities.

The prevalence estimates are also critical of the sampling methods. The sampling method, the most common in the studies reviewed, convenient sampling, could potentially contain selection bias, which either deflates or inflates the actual rate of prevalence. The random sampling methods had more credible estimates, but they were limited in their number. Also, the university type, i.e., whether it is a public or a private one, seems to play a role in the incidence of depression, with slightly higher rates of depression being reported among students of state-run colleges. This difference can be attributed to a lack of resources, overcrowding in classes and more academic and financial pressure that students may be exposed to in the public sector universities. Interestingly, subgroup analysis showed that non-medical students in Pakistan depicted a

greater prevalence of depression (60.33%) compared to the medical students (48.72%). This trend is contrasted with other international research, like Mirza et al. (2021), where medical students are usually more depressed with high levels of clinical exposure, academic rivalry, and emotional fatigue. In Pakistan, non-medical students can be seen to feel more uncertain about career opportunities, lack of planned academic opportunities and fewer institutional support mechanisms, which could intensify psychological distress. In Pakistan, the medical students tend to enjoy more defined career paths, better peer connections, and institutional support, which could be seen as protective factors. Another reason can be associated with the fact that different cultural validity and sensitivity of the tools of screening, since most of them are more frequently validated on medical student groups, which may be more accurate in the representation of distress in different academic fields.

Besides methodological and regional variables, potential publication bias and heterogeneity were also evaluated during the analysis. The test performed by Egger showed that the funnel plot is asymmetrical ($p$ = 0.019), which is a sign of the possibility of publication bias, but the Begg test was not significant. In order to assess the degree of bias, a trim-and-fill analysis was performed, and no further studies were imputed, and the adjusted pooled prevalence (50.6%; 95% CI: 47.054.2) was the same as the initial value. This consistency indicates that there was no significant publication bias on the pooled outcome. However, the large heterogeneity ($I^2$ = 98.4) shows that there is a wide range of variation between studies, which is probably caused by the differences in sample characteristics, instruments and study design. These results highlight the necessity to make a careful interpretation of the pooled prevalence and also support the significance of the standardization of methods used in future studies.

The institutional and geographical differences, along with the diversity of assessment instruments and sampling procedures, point to the necessity of culturally sensitive and methodologically comparable research methods. Regular application of the proven screening instruments in the studies would assist in drawing more valid comparisons and assist the policy makers in designing customized interventions. This great level of heterogeneity also means that it is probable that the prevalence of depression is due to the complex interaction between individual, academic, familial, and societal factors. The results are consistent with the literature on the topic in other countries, indicating that university students are particularly vulnerable to depression due to the stress of studying, social isolation, economic and change stressors (Mulaudzi, 2023; Liu et al., 2022).

In addition to the methodology, a number of other studies also highlight the need to discuss depression in university students in Pakistan. Specifically, Broton et al. (2022) discovered that high academic stress, a lack of social support, and financial insecurity had a higher predisposition toward depression among students, which is also identified in other studies that indicate that depression among students is a local issue requiring homogeneous management practices.

A theoretical interpretation of these findings can be guided by the Social Ecological Model (Bronfenbrenner, 1979; McLeroy et al., 1988), which conceptualizes mental health outcomes as the result of interacting individual, interpersonal, institutional and societal influences. From this perspective, depression among university students may emerge from the combined effects of academic pressure, family expectations, peer dynamics and sociocultural stigma. Although such a model can provide a valuable multilevel perspective of knowledge about the influences on psychological distress in a broader context, it is not the sole possible interpretative framework. Psychodynamic and other intrapersonal theories place a lot of focus on individual-level vulnerabilities, which can also contribute significantly to the susceptibility of students to depression. The identification of these various theoretical approaches aids in not being one-sided in articulating the complexity of mental health among this group of people.

Pakistan has a cultural environment which is a vital determinant of the manifestations, acknowledgment of depression as well as its administration among students of higher educational institutions. An obstacle to becoming an open society can be the prevalence of mental illness in society and the stigmatization of individuals who are subjected to psychological distress. Some of the factors identified as culturally related and which emphasize endurance and resilience among students can lead to students internalizing symptoms rather than reporting them, thereby contributing to the hidden cost of depression. Moreover, gender roles, family needs, and collectivism may also lead to a rise in mental health problems, especially among female students who have little freedom to decide on what they want to do. These cultural dimensions are significant in the development of interventions that are evidence-based as well as culturally congruent, which would reduce stigmatization and increase uptake of mental health services.

The implications of these findings are very profound. The effects of depression among university students are not only on their academic performance but also on long-term effects of how they socially operate, their employability, and quality of life. Lim et al. (2018) determined in a longitudinal study in South Korea that untreated depressive symptoms in a sample of university students predicted the likelihood of reporting chronic mental health issues and poor work outcomes, and reduced socialization during adulthood. The same patterns might be predicted in Pakistan, where there are few mental health centers, as well as the prevalence of stigma related to mental illness. It is even more problematic with the lack of mental health professionals, as there are few psychiatrists per million population, and the majority of students lack proper mental health care (Choudhry et al., 2023). The absence of mental health literacy also exacerbates the issue, as learners and teachers tend to attribute depressive symptoms to a temporary stressful experience or inner incompetence and postpone the early detection and help-seeking.

Considering these findings, it can be observed that much work must be done to address mental health problems among Pakistani students. Universities should be able to think of establishing readily available mental health services that are culturally sensitive and inclusive of counseling, mental education and stress management intervention programs. The screening programs incorporated in the academic institutions would enable the timely identification and delivery of required intervention to the at-risk students. Specifically, the idea of a peer support system and awareness campaigns will help to offer a supportive environment to students, which will result in the lower stigma in support of mental well-being. Moreover, regulatory authorities must also take into consideration the factor of mental health as a crucial component of educational policy and devote the required resources. The effectiveness of the multi-level interventions, as well as the combination of individual counseling, family interventions, and systematic policy changes, is recommended by various international studies as the most efficient in terms of the management of depression among students, and, consequently, its decrease.

Moreover, longitudinal and interventional studies ought to be considered in the future to evaluate the effectiveness of culturally modified mental health programs. The standardized testing, use of

a large sample and rural population inclusion will increase the generalizability of the findings. Furthermore, the partnership among the government, schools and mental health agencies may establish a comprehensive strategy that will help to eliminate depression and encourage resilience among students.

## Conclusion

The depression among Pakistani university students is a significant problem, whose a prevalence is pooled at about 51%. The degree of heterogeneity of the studies is high due to the variety of methodological and contextual factors, but the overall message is obvious: depression is a serious social health problem that needs to be addressed urgently and on a long-term basis. To resolve this mental health crisis, there is a need to have a multi-faceted approach that includes policy changes, resource mobilization and allocation, culturally sensitive interventions, and continuous research. Pakistan can build a healthier and more productive future generation that can overcome academic and social challenges by prioritizing the mental health of students.

**Open peer review.** To view the open peer review materials for this article, please visit http://doi.org/10.1017/gmh.2025.10125.

**Data availability statement.** Data are available from the corresponding author upon reasonable request.

**Author contribution.** YN was part of developing the concept, data curation, investigation, formal analysis, and drafting the original manuscript. MM was the supervisor of the study and participated in the investigation, formal analysis, and review and editing of the manuscript. MFA was involved in the area of investigation, formal analysis, and manuscript review and editing. SMYF helped in rewriting and editing the manuscript. The final version was reviewed by and approved by all authors, who take the responsibility to be responsible of the whole work.

**Financial support.** No individual grant was provided to support this research by any funding organization, commercial or non-profit.

**Competing interests.** The authors do not have any conflicts of interests.

**Ethics statement.** This study did not need ethical approval since it is founded on preexisting published information.

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
