## [Reviewer Report]

This is a well written systematic review summarizing the epidemiology and prevalence of depression in university students in Pakistan. Authors have used PRISMA methodology to evaluate 35 studies with over 11,000 subjects, and provide robust tables and graphs for the readers to be able to draw their own conclusions. The paper concludes that depression is a significant issue in this population. The journal being Global Mental Health, this paper makes an important contribution to global mental health awareness.

---

## [Reviewer Report]

1. If possible limit references to those published within the last 10 years.

2. Lines 21-26: Discuss if you encountered disagreements and describe in detail in what aspects did you disagree. Did you involve a third reviewer? If so, discuss his/her qualification or inclusion criteria.

3. Strengthen the theoretical underpinnings of the study.

4. Discuss the cultural implications of the results of the study.

---

## [Reviewer Report]

Strengths of the Manuscript

The systematic review and meta-analysis address a critical gap in understanding depression prevalence among Pakistani university students, a population with unique sociocultural stressors and limited mental health infrastructure. The study adheres to PRISMA guidelines, synthesizes data from 35 studies (over 11,000 participants), and employs rigorous methods (e.g., random-effects model for heterogeneity, JBI risk-of-bias assessment). Subgroup analyses (by assessment tool, sampling method, university type) provide nuanced insights, and the discussion effectively links findings to practical interventions (e.g., university counseling centers, anti-stigma campaigns), enhancing the work’s public health relevance.

2. Major Revision Suggestions

2.1 Methodology: Clarify Sampling and Assessment Tool Details

The manuscript mentions “convenience sampling (34.3%), random sampling (31.4%),” but lacks clarity on how “random sampling” was operationalized (e.g., simple random, stratified random) across studies. This ambiguity limits the interpretation of selection bias, as different randomization methods vary in representativeness. Please specify the type of random sampling used in included studies and whether this variation was accounted for in subgroup analyses.

- While the DASS-21 is noted as the most common assessment tool (40% of studies), the manuscript does not address whether the cut-off scores for depressive symptoms were consistent across studies. For example, some studies may have used a DASS-21 depression subscale score ≥10, while others used ≥14—this inconsistency could drive heterogeneity. Please report the range of cut-off scores for key tools (DASS-21, PHQ-9, AKUADS) and analyze whether cut-off variation contributes to prevalence differences in subgroup or meta-regression analyses.

2.2 Results: Address Publication Bias and Heterogeneity Interpretation

Egger’s test indicates significant publication bias (p=0.0185), but the discussion only notes this “possibility” without proposing strategies to mitigate its impact (e.g., trim-and-fill analysis, sensitivity analysis excluding small studies with extreme prevalence rates). Please conduct and report a trim-and-fill analysis to adjust the pooled prevalence estimate and discuss how publication bias might alter the conclusion that “~51% of students experience depressive symptoms.”

The high heterogeneity (I²>94% across subgroups) is attributed to “assessment instruments, sampling techniques, regional differences,” but meta-regression analyses to quantify the contribution of specific factors (e.g., year of study, city income level) are absent. Meta-regression could identify which variables most strongly drive prevalence variation (e.g., whether post-2020 studies with validated tools show lower heterogeneity). Please add a meta-regression table analyzing the association between pre-specified covariates (e.g., study year, sample size, tool validity) and depression prevalence.

2.3 Discussion: Strengthen Contextual and Comparative Analysis

The discussion compares Pakistani students’ depression rates to “global trends” but lacks specific benchmarks (e.g., prevalence among university students in other South Asian countries like India or Bangladesh, or global averages from WHO reports). Adding such comparisons would contextualize whether Pakistan’s 51% pooled prevalence is an outlier or consistent with regional/low-middle-income country patterns. For example, cite recent meta-analyses (e.g., Mekonnen et al., 2024 on African medical students) to frame the findings.

- The manuscript notes that “non-medical students exhibited higher prevalence (60.33%) than medical students (48.72%),” which contradicts some international studies (e.g., Mirza et al., 2021, which reports higher rates in medical students). The discussion does not explore potential reasons for this discrepancy (e.g., whether non-medical students in Pakistan face unique stressors like limited career prospects, or whether assessment tools are less validated for non-medical populations). Please elaborate on this finding, linking it to sociocultural or educational context in Pakistan.

2.4 Tables and Figures: Improve Clarity and Completeness

Table 1 (“Basic characteristics of included studies”) is partially incomplete: some rows (e.g., Bukhari & Khanam, 2015; Lail et al., 2021) have missing values for “Age (Mean) years” or “Prevalence (n).” Additionally, the “Medical or nonmedical” column uses “Both” without specifying the proportion of medical vs. non-medical students in mixed cohorts. Please complete missing data and clarify the composition of mixed samples to enable transparent assessment of subgroup differences.

- Figure 2 (“Meta-analysis of prevalence”) and Figure 3 (“Funnel plot”) are referenced but lack detailed captions (e.g., Figure 2 should specify whether it presents fixed vs. random-effects estimates; Figure 3 should note the axes labels and whether points represent individual studies). Please revise captions to include this information, ensuring compliance with journal guidelines for figure reproducibility.

3. Minor Revision Suggestions

3.1 Abstract and Keywords

The abstract lists “Radiography and Medical Imaging Technology” as a topic, which is irrelevant to the study’s focus on depression in university students. Please remove this topic to align with the manuscript’s content.

- Keywords include “Academic stress” and “Mental illness prevention,” but the manuscript does not analyze “academic stress” as a predictor (e.g., meta-regression of stress levels vs. depression prevalence) or evaluate existing prevention programs. Please either revise keywords to reflect analyzed constructs (e.g., “DASS-21” or “convenience sampling”) or add a brief section in the discussion on how academic stress interacts with other factors (e.g., financial difficulties) to influence depression.

3.2 Introduction: Refine Depression Definition and Burden

-The introduction defines depression using symptoms (e.g., “persistent low mood, fatigue”) but does not explicitly link these to DSM-5 or ICD-11 criteria, which are standard in global mental health research. Please integrate DSM-5/ICD-11 diagnostic criteria to ensure consistency with international definitions.

- The claim that depression is “the major cause of disability worldwide” cites Ferrari et al. (2010) and Whiteford et al. (2013), but more recent data (e.g., Global Burden of Disease 2019 or 2021) are available. Updating these citations would strengthen the manuscript’s currency.

overall:

The manuscript makes a valuable contribution to global mental health research on student populations. Addressing the above issues—particularly clarifying sampling/assessment tool details, mitigating publication bias, and contextualizing findings—will enhance its methodological rigor and impact. With revisions, this work will serve as a critical resource for policymakers designing mental health interventions for Pakistani university students.

---

## [Reviewer Report]

Dear Authors,

I find your paper highly scholarly, well-written, and conceptually strong. The structure and clarity of your arguments demonstrate rigorous academic work. However, I have a few suggestions to further strengthen the manuscript and enhance its potential for publication:

1. Inclusion and Exclusion Criteria

Please provide a more detailed discussion of the study’s inclusion and exclusion criteria. A clear and explicit description will help readers fully understand how participants or sources were selected and will strengthen the methodological transparency of the manuscript.

2. Discussion of Assessment Instruments

In the Discussion section, kindly expand your analysis of the various assessment instruments used in the study. Elaborate on their implications, strengths, limitations, and how these may have influenced the results. This will contribute to a deeper and more critical interpretation of the findings.

3. Lines 38–46: Strength of Theoretical Claims

The statements in this section are particularly strong, especially the sentence: “Theoretically, the results can be explained by the Social Ecological Model… and not the vulnerability of the individuals.” I suggest toning this down, as psychodynamic theories also offer substantial explanations for individual propensity to depression. To maintain theoretical balance, you may consider adding this point to the scope and limitations of the study, acknowledging that alternative theoretical frameworks may also account for the observed outcomes.